



# On the link between Earth tides and volcanic degassing

Florian Dinger[1,2], Stefan Bredemeyer[3,4], Santiago Arellano[5], Nicole Bobrowski[1,2], Ulrich Platt[1,2], and Thomas Wagner[1]

[1]Max-Planck Institute for Chemistry, Mainz, Germany
[2]Institute of Environmental Physics, University of Heidelberg, Germany
[3]GEOMAR, Kiel, Germany
[4]GFZ, Potsdam, Germany
[5]Department of Space, Earth and Environment, Chalmers University of Technology, Gothenburg, Sweden

*Correspondence to:* Florian Dinger (fdinger@iup.uni-heidelberg.de)

**Long-term measurements of volcanic gas emissions conducted during the recent decade suggest that under certain conditions the magnitude or chemical composition of volcanic emissions exhibits periodic variations with a period of about two weeks. A possible cause of such a periodicity can be attributed to the Earth tidal potential. The phenomenology of such a link has been debated for long, but no quantitative model has yet been proposed. The aim of this paper is to**

**elucidate whether a causal link from the tidal forcing to variation in the volcanic degassing can be traced analytically. We model the response of a simplified magmatic system to the local tidal gravity variations and derive a periodical vertical magma displacement in the conduit with an amplitude of 0.1-1 m, depending on geometry and physical state of the magmatic system. We find that while the tide-induced vertical magma displacement has presumably no significant direct effect on the volatile solubility, the differential magma flow across the radial conduit profile may result in a sig-**

**nificant increase of the bubble coalescence rate in a depth of several kilometres by up to several ten percent. Because bubble coalescence facilitates separation of gas from magma and thus enhances volatile degassing, we argue that the derived tidal variation may propagate to a manifestation of varying volcanic degassing behaviour. The presented model provides a first basic framework which establishes an analytical understanding of the link between the Earth tides and volcanic degassing.**

## 1 Introduction

Residual gravitational forces of the Moon and the Sun deform the Earth's surface and interior periodically and thus lead to the so-called Earth tides. The tidal potential can be modelled as the result of the interference of an infinite number of sinusoidal tidal harmonics with precisely known frequencies and amplitudes (Darwin, 1883; Doodson, 1921). At the equator, the tidal potential varies predominantly with a semi-diurnal periodicity. The amplitude of the semi-diurnal cycle is modulated within

the so-called spring-neap tide cycle with a periodicity of 14.8 days caused by the interference of the lunar semi-diurnal tide and the solar semi-diurnal tide. The peak-to-peak amplitude of the associated semi-diurnal gravity variations is $a_{astro}^{st} = 2.4\,\mu\mathrm{m\,s^{-2}}$ during spring tide, $a_{astro}^{nt} = 0.9\,\mu\mathrm{m\,s^{-2}}$ during neap tide, and at intermediate level at other times of the cycle. At mid latitudes, the tidal potential varies predominantly with diurnal periodicity and at other latitudes both periodicities mix. The spring-neap



tide cycle is however manifested everywhere and with maximum variability at the equator (Agnew, 2007). The tidal potential firstly gives rise to a periodical elevation of the Earth's crust with a semi-diurnal peak-to-peak variation of up to about $50\,\mathrm{cm}$ (maximum at the equator), and secondly all crustal compartments exhibit an additional semi-diurnal gravity variation by up to $1.16 \cdot a_{astro}^{st}$ (Harrison et al., 1963; Baker, 1984). This gravity variation typically has no effect on the rigid solid crust but can

cause fluid movement, e.g. prominently manifested in the form of the ocean tides (Ponchaut et al., 2001).

Evidences for tidal impacts on volcanism have been gathered by numerous empirical studies which detected a temporal proximity between tidal extrema and volcanic eruptions (Johnston and Mauk, 1972; Hamilton, 1973; Dzurisin, 1980) or seismic events (McNutt and Beavan, 1981, 1984; Ide et al., 2016; Petrosino et al., 2018), or found a correlation between the spring-neap tide cycle and variations in volcanic deformation (De Mendoca Dias, 1962; Berrino and Corrado, 1991) or variations in the

volcanic gas emissions.

The tide-induced stress variations ($\sim 0.1 - 10\,\mathrm{kPa}$) appear to be negligibly small in comparison to tectonic stresses ($\sim 1 - 100\,\mathrm{MPa}$) or stresses caused by pressure and temperature gradients within a shallow magmatic system ($\sim 1\,\mathrm{MPa}$). The rate of tidal stress change can, however, be around $1\,\mathrm{kPa\,h^{-1}}$ and thus potentially exceeds stress rates of the other processes by one to two orders of magnitude (Sparks, 1981; Emter, 1997; Sottili et al., 2007). Furthermore, these subtle stress

variations may cause an amplified volcanic reaction, when the tidal variations cause, e.g., a widening of tectonic structures (Patanè et al., 1994), a periodic decompression of the host rock (Sottili et al., 2007; Sottili and Palladino, 2012), self-sealing of hydrothermal fractures (Cigolini et al., 2009), or a mechanical excitation of the uppermost magmatic gas phase (Girona et al., 2018).

First studies on the co-variations of tidal pattern and volcanic gas emissions hypothesised about a possible tidal impact on the

observed sulphur dioxide ($SO_2$) emission fluxes at Masaya (Stoiber et al., 1986) and Kilauea (Connor et al., 1988). Since the 2000s, automatic scanning networks based on UV-spectrometers (e.g. Galle et al., 2010) provide multi-year time series of volcanic gas emissions of $SO_2$ and bromine monoxide (BrO). The availability of such data sets enabled extensive investigation of long-term degassing variations. Correlation with the tidal long-term patterns have been reported for the $SO_2$ emission fluxes of Villarrica and Llaima (Bredemeyer and Hansteen, 2014), and the $BrO/SO_2$ molar ratios in the gas plume of Cotopaxi (Dinger

et al., 2018). Another possible but less significant correlation has been reported for the $SO_2$ emission fluxes of Turrialba (with a periodicity somewhere between 9.1 days and 16.7 days, Conde et al., 2014). Furthermore, Lopez et al. (2013) reported a periodicity of roughly 16 days in the $SO_2$ emission fluxes of Redoubt retrieved from the satellite-based OMI-instrument (the authors proposed that this periodicity is however an artefact of the satellite orbit rather than a tidal signal). In addition, correlation with the tidal long-term patterns have been reported for the diffuse Radon degassing of Terceira (Aumento, 2002) and

Stromboli (Cigolini et al., 2009).

Cycles in volcanic degassing patterns are not unique to periodicities which match the tidal potential. Many studies reported periodic volcanic degassing pattern with periods of minutes (e.g. Fischer et al., 2002; Boichu et al., 2010; Campion et al., 2012, 2018; Tamburello et al., 2013; Pering et al., 2014; Ilanko et al., 2015; Moussallam et al., 2017; Bani et al., 2017). In contrast, observations of long-term periodicities are rare. Besides the above mentioned about biweekly periodicities, periodic long-term

pattern with periodicities of 50 days and 55 days have been observed in the $SO_2$ emission flux of Soufrière Hills (Nicholson



et al., 2013) and Plosky Tolbachik (Telling et al., 2015), respectively.

In the view of the growing number of studies revealing similar biweekly patterns in volcanic activity, this paper investigates whether a causality from the tidal potential to variations in the volcanic degassing is analytically traceable in a comprehensible way. High temperature gas emissions of persistently strong passively degassing volcanic systems are commonly thought being

fed by sustained magma convection reaching the uppermost portions of the volcanic conduit, where volatile-rich low-viscous magma ascends through essentially degassed magma of higher viscosity, which in turn descends at the outer annulus of the conduit (Kazahaya et al., 1994; Palma et al., 2011; Beckett et al., 2014). Magma ascent rates associated to such convective flow typically vary roughly between $1 - 100\,\mathrm{m\,h^{-1}}$ (Cassidy et al., 2015, 2018) and thus are orders of magnitudes larger than what we can derive for potentially tide-induced vertical magma displacement rates of at most $0.6\,\mathrm{m}$ within $6\,\mathrm{h}$ (if not further

amplified). A comprehensive model of the tidal impact on the magma motion thus requires a coupling of the convective and the tide-induced transport mechanisms.

Our conceptual model aims to provide the first step by investigating the purely tide-induced transport mechanism acting on the low-viscous inner magma column neglecting any interferences between the magma ascent and the tidal mechanism, i.e. the model ignores the magma convection in the column. We model the response of such a quasi-static magmatic system (volcanic

conduit connected to a laterally more extended deeper magma reservoir) to tide-induced gravity variations analogously to the response of a classical mercury thermometer to temperature variations: the tide drives a periodical expansion of the magma in the reservoir which leads to a periodical vertical displacement of the low-viscous magma column in the conduit.

We derive the temporal evolution and amplitude of the vertical magma displacement across the radial conduit profile and examine its impact on the bubble coalescence rate. In order to introduce our novel approach comprehensibly, the modelled processes

and conditions are as simplified as suitable; the major simplifications are listed in Appendix A. All findings in this paper are derived analytically. For illustration of the model output we nevertheless present some quantitative estimates for two exemplary magmatic systems. These examples are intended to match simplified versions of mid-latitude Villarrica and equatorial Cotopaxi volcanoes where covariation between outgassing activity and Earth tidal movements has been observed previously (Bredemeyer and Hansteen, 2014; Dinger et al., 2018). The associated model parameter sets are listed in Table 1. Further, all

quantitative estimates are presented for the spring tide, and the consequences of the contrast between spring tide and neap tide are discussed in the last part of this paper.

## 2   Tide-induced magma displacement in the conduit

### 2.1   Model set-up

We model the magmatic system analogously to established convection models (Kazahaya et al., 1994; Palma et al., 2011;

Beckett et al., 2014), with the exception that the descending high-viscous magma annulus is assumed to be not affected by the tide-induced dynamics and therefore is considered as an effective part of the host rock, while "conduit" refers in our model exclusively to the ascending low-viscous magma column. We assume the conduit to be a vertically oriented cylinder with length $L_c$, radius $R_c$, and cross-sectional area $A_c = \pi \cdot R_c^2$ which is confined by the penetrated host rock (and high-viscous




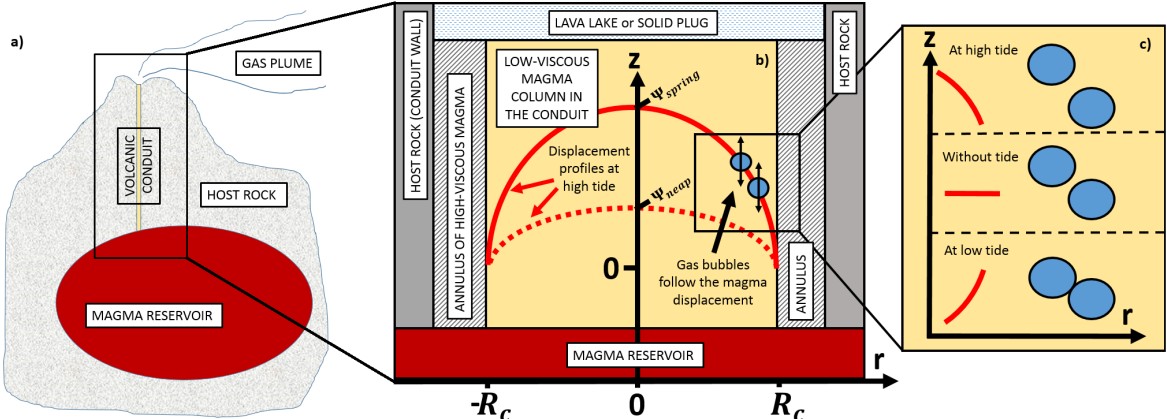

**Figure 1. a)+b)** Sketch of the model set-up. The model compartments are noted by white boxes and depicted not to scale. **b)** The semi-diurnal tide causes a radial magma displacement profile in the conduit with different amplitudes during spring tide and neap tide, respectively. **c)** Concept of the tide-enhanced bubble coalescence: Two bubbles which are initially close to each other (see "without tide") exhibit differential vertical tide-induced displacements what enhances the chance for bubble coalescence (here "at low tide").

magma annulus), connected to a deeper, laterally more extended magma reservoir with volume $V_r$ and centre of mass at a depth $D_r$, and either exhibiting an open vent or capped by a gas-permeable solid plug (Figure 1). The magmatic melt in the conduit is modelled as a mixture of a liquid phase and a gas phase having a mean density $\rho_{melt}$, which varies with pressure and thus depth, a constant kinematic bulk viscosity $\nu$, and homogeneous local flow properties. The magma compressibility $\beta(\phi)$

strongly depends on the gas volume fraction $\phi$ and lies between the compressibility $\beta(0) = 2 \cdot 10^{-10}\,\mathrm{Pa}^{-1}$ of volatile-rich rock and the compressibility $\beta_0(1) \approx p^{-1}$ of an ideal gas (see e.g. Tripoli et al., 2016). The magmatic melt in the reservoir is modelled to be volatile-rich but hosting no gas phase of significant volume and thus having a constant compressibility $\beta_r \approx \beta(0)$. Further, the quasi-static condition implies a steady-state density stratification within the magma and also with respect to the host rock (no neutral buoyancy, Parfitt et al., 1993). In this equilibrium, we assume a constant hydrostatic

pressure gradient $(\nabla p)_{vert}$.

## 2.2 Response of the host rock on tidal stresses

Magma pathways are often located at intersection points of large-scale fault systems (Nakamura, 1977; Takada, 1994), or respectively in fault transfer zones (e.g., Gibbs, 1990), where the surrounding host rock geometry is relatively sensitive to

directional changes in pressure. The vertical and horizontal components of the tidal force exert additive shear tension on the host rock, potentially causing a compression of the host rock (Sottili et al., 2007) or a differential slip between both sides of the fault system (Ide et al., 2016). Both mechanisms can cause an increase in the areal conduit cross section. Connected to the magma reservoir, such an increasing conduit volume is accompanied by decompression and thus causes a magma to flow



from the reservoir to the conduit which pushes the initial magma column in the conduit upwards until the initial hydrostatic pressure gradient is re-established. Vice versa a relative decrease in the areal conduit cross section leads to an effective descent of the initial magma column in the conduit. For a given periodic area increase $\Delta A_c$, the amplitude $\Delta z_{hr}$ of this additive elevation-descent cycle of the centre of mass of the initial magma column is given by

$$5 \quad \Delta z_{hr} = \frac{L_c}{2} \cdot \frac{\Delta A_c}{A_c + \Delta A_c} \approx \frac{L_c}{2} \cdot \frac{\Delta A_c}{A_c} \tag{1}$$

The quantitative scale of tide-induced conduit cross section variations is presumably hardly accessible. The theoretical horizontal components of the tide-induced ground surface displacement are up to about $\pm 7\,\text{cm}$ (Baker, 1984). Slip-induced dilation of faults with widths in the sub-centimetre range thus appear to be plausible. For illustration, a conduit radius increase by $\Delta R_c = 1\,\text{mm}$ would result in an additive vertical centre of mass displacement by $\Delta z_{hr} = 0.33\,\text{m}$ for Villarrica and $\Delta z_{hr} = 0.13\,\text{m}$ for Cotopaxi. As a remark, these mechanisms do not require a cylindrical conduit and fault-slip mechanisms would rather lead to an uni-directional area increase rather than a homogeneous radial increase.

### 2.3 Tide-induced magma expansion in the reservoir

The semi-diurnal tide causes a sinusoidal variation of the gravitational acceleration with angular frequency $\omega_{sd} = 1.5 \cdot 10^{-4}\,\text{rad}\,\text{s}^{-1}$ and amplitude (equals the half peak-to-peak amplitude) $a_0^{st} = 1.4\,\mu\text{m}\,\text{s}^{-2}$ during spring tide and $a_0^{nt} = 0.5\,\mu\text{m}\,\text{s}^{-2}$ during neap tide. Besides those host rock mechanisms triggered by the tidal stresses, also these tide-induced gravity variations may cause a periodical elevation of the magma in the inner conduit.

The compressible magma in the reservoir is pressurised by the hydrostatic load whose weight is proportional to the local gravitational acceleration $g$. A reduction of the local gravitational acceleration by $a_0$ leads to a decompression and thus expansion of the magma in the reservoir by $\Delta V_r = \frac{a_0}{g} \cdot (\nabla p)_{vert} \cdot D_r \cdot \beta_r \cdot V_r$. The tidal force can accordingly lead to a periodical magma expansion-shrinkage cycle in the reservoir with a semi-diurnal periodicity and an amplitude modulation within the spring-neap tidal cycle of up to $\Delta V_r \sim \mathcal{O}(100 - 1000\,\text{m}^3)$.

The realisation of this additional magma volume implies a displacement and thus compression of the host rock at the location of maximum host rock compressibility. This is typically true for the conduit. Assuming that the magma expansion in the reservoir ultimately and exclusively causes an increase of the conduit volume, the volume increase causes an elevation of the centre of mass of the initial magma column in the conduit by

$$\Delta z_{dec} = \frac{\Delta V_r}{A_c} = \frac{a_0}{g} \cdot (\nabla p)_{vert} \cdot D_r \cdot \beta_r \cdot \frac{V_r}{\pi \cdot R_c^2} \tag{2}$$

In the general case, the additional volume could be realised by a slight increase of the conduit radius by $\Delta R_{dec} \approx \frac{R_c}{2} \cdot \frac{\Delta z_{dec}}{L_c} \sim \mathcal{O}(1\,\text{mm})$ caused, e.g., by the tidal stresses. If the magmatic system has an open vent, the additional volume can alternatively be realised by an elevation of the lava lake level and thus without a host rock compression.

The tide-induced gravity variations result analogously in an expansion of the initial magma column in the conduit. This effect is however typically negligible compared to the reservoir-effect for sufficiently large reservoirs (volume contrast between reservoir and conduit of more than 1000, see Table 1), we thus neglect for simplicity the effect of the expansion of the initial





magma column in the conduit.

The responses of the overall magmatic system on the tidal stresses and tide-induced gravity variations act simultaneously and in phase with the tidal force. The overall vertical tide-induced magma displacement in the conduit $\Delta z_{max}$ can thus by larger then the individual mechanisms, i.e. $\{\Delta z_{hr}, \Delta z_{dec}\} \leq \Delta z_{max} < \Delta z_{hr} + \Delta z_{dec}$. In the following we focus on the reservoir expansion mechanism only in order to keep the derivation of the model parameters strictly analytical. The host rock mechanism is therefore reduced to establishing the required areal conduit cross section increase by $\Delta R_{dec}$.

**Table 1.** Choice of model parameters, motivated by conditions at (1) Villarrica volcano located at $39.5°$S hosting a persistent lava lake of basaltic composition; and (2) Cotopaxi volcano located at $0.7°$S which preferentially erupts andesitic magma and intermittently is capped by a solid plug. If not stated otherwise, all numerical values in this manuscript are calculated with these parameters.

| Model parameter | | | Location-independent constants/assumptions | | | |
|---|---|---|---|---|---|---|
| Physical Parameter | Notation | Unit | value | literature | | |
| pure spring tide amplitude | $a_0^{st}$ | $\mathrm{m\,s^{-2}}$ | $1.4 \cdot 10^{-6}$ | Baker (1984), at the equator | | |
| semi-diurnal periodicity | $\omega_{sd}$ | $\mathrm{rad\,s^{-1}}$ | $1.5 \cdot 10^{-4}$ | Baker (1984) | | |
| hydrostatic pressure gradient | $(\nabla p)_{vert}$ | $\mathrm{Pa\,m^{-1}}$ | $2.7 \cdot 10^{4}$ | for andesitic host rock | | |
| solubility coefficient of water | $K_{H_2O}$ | $\mathrm{Pa^{-1}}$ | $1 \cdot 10^{-11}$ | Liu et al. (2005) | | |
| magma compressibility | $\beta_r$ | $\mathrm{Pa^{-1}}$ | $2 \cdot 10^{-10}$ | for the magma in the deep reservoir, see Appendix B | | |
| (local) gas volume fraction | $\phi$ | | 0.2 | chosen reference value | | |
| | | | **Villarrica** | | **Cotopaxi** | |
| conduit length | $L_c$ | km | 2 | see Appendix B | 4 | see Appendix B |
| conduit radius | $R_c$ | m | 6 | see Appendix B | 40 | see Appendix B |
| reservoir volume | $V_r$ | $\mathrm{km^3}$ | 35 | see Appendix B | 35 | see Appendix B |
| depth of reservoir (c.o.m.) | $D_r$ | km | 3 | see Appendix B | 8 | see Appendix B |
| kinematic viscosity | $\nu$ | $\mathrm{m^2\,s^{-1}}$ | 0.1 | Palma et al. (2011) | 4 | (andesitic melt) |
| melt density | $\rho_{melt}$ | $\mathrm{kg\,m^{-3}}$ | 2600 | Palma et al. (2011) | 2500 | (andesitic melt) |
| melt weight fraction of water | $C_{H_2O}^0$ | % | 2 | Palma et al. (2011) | 5 | Martel et al. (2018) |
| max. vertical tidal acceleration | $a_0$ | $\mathrm{m\,s^{-1}}$ | $0.61 \cdot a_0^{st}$ | Baker (1984), @$39.5°$S | $a_0^{st}$ | Baker (1984), @$0.7°$S |
| gravitational acceleration | $g$ | $\mathrm{m\,s^{-2}}$ | 9.81 | @$39.5°$S | 9.78 | @$0.7°$S |
| magma temperature | T | °C | 1200 | | 1000 | |

## 2.4 Radial flow profile in the conduit

The tide-induced vertical magma displacement in the conduit is delayed and extenuated by a viscosity-induced drag force. We access the temporal evolution and amplitude of the tide-induced displacement via the force (per unit mass) balance acting on



the centre of mass of the magma column in the conduit

$$
\underbrace{\gamma \cdot \dot{z}(t)}_{\text{drag force}} = \overbrace{\underbrace{a_0 \cdot \sin(\omega_{sd} \cdot t)}_{\text{tidal force}} - \underbrace{\omega_0^2 \cdot z(t)}_{\text{restoring force}} - \underbrace{\ddot{z}(t)}_{\text{inertial force}}}^{\text{external force}}
\qquad (3)
$$

where the two model parameters are the bulk damping rate $\gamma$ and the eigenfrequency $\omega_0$ of the magma column. The restoring force ensures that the centre of mass displacement tends to the current "equilibrium" displacement associated to the current strength of the tidal force, i.e. $a_0 = \omega_0^2 \cdot \Delta z_{max}$. We further assume a Newtonian bulk drag force proportional to the flow velocity.

The continuity condition implies that the magma flows faster in the conduit centre than close to the boundary between the low-viscous and high-viscous magma/host rock. Accordingly, we assume a no-slip condition at the conduit boundary $r = R_c$ and derive the analytical solution of the tide-induced parabolic vertical displacement profile $z(r,t)$ in the conduit

$$
\begin{cases}
z(r,t) & = \Psi \cdot \left[1 - \left(\dfrac{r}{R_c}\right)^2\right] \cdot \sin(\omega_{sd} \cdot t - \varphi_0) \\[2mm]
\Psi & = \dfrac{2 \cdot a_0}{\sqrt{(\omega_0^2 - \omega_{sd}^2)^2 + (\gamma \cdot \omega_{sd})^2}} \\[2mm]
\varphi_0 & = \arctan\left(\dfrac{\gamma \cdot \omega_{sd}}{\omega_0^2 - \omega_{sd}^2}\right) \\[2mm]
\gamma & = \dfrac{8 \cdot \nu}{R_c^2} \\[2mm]
\omega_0^2 & = \dfrac{a_0}{\Delta z_{dec}} = \dfrac{g \cdot \pi \cdot R_c^2}{\beta_r \cdot V_r \cdot D_r \cdot (\nabla p)_{vert}}
\end{cases}
\qquad (4)
$$

with the radial coordinate $0 \leq r \leq R_c$, the maximum vertical magma displacement amplitude $\Psi$ (which equals twice the centre of mass displacement) and the phase shift $\varphi_0$ between tidal force and magma displacement in the conduit (see Appendix C). For Villarrica, the model implies a tidal displacement amplitude of $\Psi_{vill}^{st} = 0.45\,\text{m}$ which lags behind the tide by $\varphi_{0,vill} \cdot \omega_{sd}^{-1} = 2.0\,\text{h}$, where the displacement is predominantly limited by drag force. For Cotopaxi, the tidal displacement amplitude is $\Psi_{coto}^{st} = 0.09\,\text{m}$ and lags by $\varphi_{0,coto} \cdot \omega_{sd}^{-1} = 0.2\,\text{h}$, where the displacement is predominantly limited by the restoring force. In comparison, the direct tide-induced gravity variations leads to a variation of the hydrostatic pressure by $10 - 100\,\text{Pa}$. In the context of the hydrostatic pressure gradient this pressure variation has a similar effect as a vertical magma displacement by about 1 mm, thus rendering the direct tidal impact negligible compared to the here derived indirect mechanism.

## 3 Tide-enhanced bubble coalescence

Integrated over a semi-diurnal cycle, the tides do not result in a net magma displacement. A link from tides to degassing thus requires tide-enhanced mechanisms which irreversibly change the state of magmatic gas phase. Bubble growth constitutes a predominantly exergonic and thus irreversible mechanism because the bubble surface tension inhibits or at least damps bubble shrinkage and dissolution (Prousevitch et al., 1993). Within a tide-induced radial displacement profile, neighbouring gas bubbles can exhibit differential tide-induced vertical displacements potentially enhancing the bubble coalescence rate (see





Figure 1c and Appendix D). The variation of the bubble coalescence rate leads to bigger bubbles and thus the tide can indeed couple to an irreversible mechanism.

In this section, we set-up a simplified formalisation of the magmatic gas phase and the typically predominant mechanisms which govern the bubble coalescence rate, and estimate the relative tide-induced enhancement of the bubble coalescence by a

comparison with these classical mechanisms. We consider a magma layer in the conduit at a particular depth, accordingly, the parameters discussed in the following describe the local conditions within a small volume of magma and should not be confused with the integrated bulk values for the total magma column. The variation of the tide-induced enhancement at different magma depths is discussed in the subsequent section.

### 3.1 Gas bubbles in magmatic melt

The dominant part of the magmatic volatile content is typically water vapour followed by carbon dioxide, sulphur compounds and minor contributions from a large number of trace gases such as halogen compounds (Oppenheimer et al., 2014). For simplicity, we assume that all macroscopic properties of the gas phase are dominated by the degassing of water vapour, in particular that the gas volume fraction $\phi$ exclusively consists of water vapour. The volatile solubility of magmatic melts is primarily pressure dependent with secondary dependencies on temperature, melt composition, and volatile speciation (Gonnermann

and Manga, 2013). The pressure-dependency of the water solubility $C_{H_2O}$ in magmatic melt is given in first approximation by $C_{H_2O}(p) = \sqrt{K_{H_2O} \cdot p}$ with the corresponding solubility coefficients $K_{H_2O}$ (find an empirical formulation in Liu et al., 2005). For the local gas volume fraction $\phi(p)$ at a depth associated with the pressure $p$, we obtain

$$\phi(p) = \frac{\rho_{melt}(p)}{\rho_{gas}(p)} \cdot \left( C_{H_2O}^0 - \sqrt{K_{H_2O} \cdot p} \right) \tag{5}$$

with the total water weight fraction $C_{H_2O}^0$ of the magmatic melt and the mass densities of the gas phase $\rho_{gas}$ and of the overall

melt (liquid + gas) $\rho_{melt}$.

The gas phase consists of separated bubbles as long as the gas volume fraction is below the percolation threshold of $\phi_{perc} = 0.3 - 0.7$ (the variation is due to the range of different magmatic conditions, Rust and Cashman, 2011). Bubbles typically vary in size following a power law (Cashman and Marsh, 1988; Blower et al., 2003) and in shape from spherical to ellipsoidal (Rust et al., 2003; Moitra et al., 2013). While models based on polydisperse bubble size distributions are available (Sahagian and

Proussevitch, 1998; Huber et al., 2013; Mancini et al., 2016), a common starting point to analyse the temporal evolution of the bubbles is nevertheless the assumption of a monodisperse size distribution of spherical bubbles (Prousevitch et al., 1993; Lensky et al., 2004).

We note the bubble size distribution $\delta_b^{size}(f \in \mathbb{R}^+)$ with respect to the bubble radius (rather than the volume), i.e. the bubble radius is given by $r_b = f \cdot R_b$ with the hypothetical bubble radius $R_b(p)$ of a monodisperse bubble size distribution. The best

estimate of the bubble size distribution is a power-law which however requires three parameters: the exponent and the lower and upper truncation cut-off (Lovejoy et al., 2004). The following analysis is conducted for an arbitrary bubble size distribution,



nevertheless, for a basic quantitative estimate, we mimic a proper power-law bubble size distribution by the simpler

$$\tilde{\delta}_b^{size}(f) = \begin{cases} 1-q & : & f = 1 \\ q & : & f = \sqrt[3]{2} \end{cases} \tag{6}$$

with $0 \leq q < \frac{1}{2}$ which represents a monodisperse distribution except for a fraction of $q$ bubbles which emerged from a past coalescence of two bubbles with $f = 1$.

## 3.2 Bubble motion and bubble coalescence

Diffusion-driven volatile degassing can only take place in the immediate vicinity of a bubble and when the supersaturation pressure is larger than the bubble surface tension (Proussevitch and Sahagian, 2005). The volatile degassing rate is thus controlled by the spatial bubble distribution as well as the bubble size distribution (Lensky et al., 2004). Both distributions change during bubble rise which is caused by a vertical ascent of the overall magma column/parcel with velocity $v_{melt}$ and a superimposed bubble buoyancy with a velocity $v_{buoy}$ which reads for a bubble with radius $r_b$ (Stoke's law)

$$v_{buoy}(r_b) = \frac{2 \cdot g \cdot r_b^2}{9 \cdot \nu} \cdot \left(1 - \frac{\rho_{gas}}{\rho_{melt}}\right) \approx \frac{2 \cdot g \cdot r_b^2}{9 \cdot \nu} \tag{7}$$

If the buoyancy velocity is negligible compared to the magma ascent, the bubble flow is called "disperse"; if the bubble buoyancy velocity contributes significantly to the overall bubble ascent, the bubble flow is called "separated" (Gonnermann and Manga, 2013). Rising bubbles grow continuously because of (1) decompression and (2) the increasing volatile degassing rate due to the associated decreases of the magmatic volatile solubility and of the bubble surface tension. Bubble coalescence accelerates the bubble growth.

Bubble coalescence requires two bubble walls to touch and ultimately to merge. Once two bubbles are sufficiently close to each other, near-field processes such as capillary and gravitational drainage cause a continuous reduction of the film thickness between the bubble walls until the bubbles merge after drainage times ranging from seconds to hours depending on the magmatic conditions (Herd and Pinkerton, 1997; Castro et al., 2012; Nguyen et al., 2013).

For small gas volume fractions, however, the initial distance between bubbles is large compared to the bubbles dimensions and the coalescence rate is dominated by bubble transport mechanisms acting on longer length scales. Because bubble diffusion is typically negligibly small, bubble walls can only approach when a particular mechanism leads to differential bubble rise velocities or by bubble growth. In magmas with a sufficiently separated bubble flow, two neighbouring bubbles of different size can approach each other vertically due to the differential buoyancy velocities (Manga and Stone, 1994; Lovejoy et al., 2004). In magmas with a disperse bubble flow, in contrast, the relative position of bubble centres remains fixed thus bubble coalescence is controlled by the bubble expansion rate caused by the ascent of the overall magma column/parcel.

## 3.3 Comparison of bubble coalescence mechanism

The proposed tide-induced bubble transport mechanism is compared in the following with the classically predominant bubble transport/approaching mechanisms in order to estimate the relative contribution of the tidal mechanism on the overall coales-



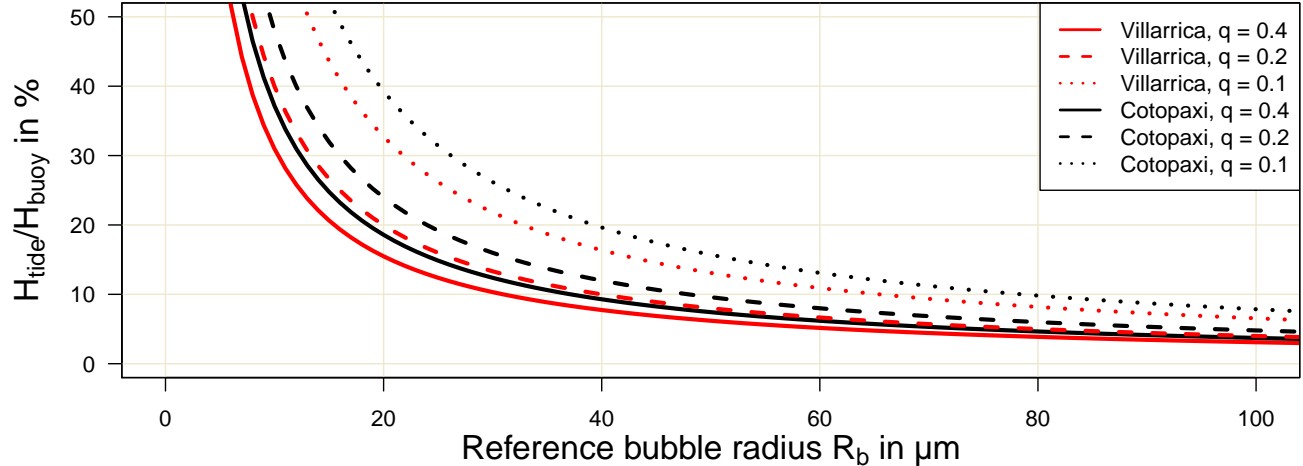

**Figure 2.** Relative contribution of the tidal mechanism (magnitude given by $H_{tide}$) on the bubble coalescence rate for a purely separated bubble flow (magnitude given by $H_{buoy}$) in dependency of the reference bubble radius $R_b$ and the degree of polydispersivity $q$. The reference bubble radius is reciprocally linked to the depth of the particular magma layer.

cence rate. We access the (absolute) strength of a particular transport mechanism by its "collision volume" $H_i$ (see Appendix D). The tidal mechanism is noted by $H_{tide}$. For comprehensibility, we focus on a comparison of the tidal mechanisms with the two "end-member" scenarios of a purely separated ($H_{buoy}$) and a purely disperse ($H_{disp}$) bubble flow, respectively. A more comprehensive formulation of the classically predominant bubble transport/approaching mechanisms can be found e.g.
in (Mancini et al., 2016).

For a separated bubble flow, the relative tidal contribution on the bubble coalescence rate depends reciprocally on the reference bubble radius $R_b$ and on the degree of polydispersivity $q$ (see Figure 2). For $q = 0.1 - 0.4$, the tidal mechanism contributes at least $10\,\%$ to the overall bubble coalescence rate for a range of reference bubble radii of $R_b = 32 - 65\,\mu$m for Villarrica and $R_b = 37 - 78\,\mu$m for Cotopaxi. For comparison, Castro et al. (2012) obtained from rhyolite decompression experiments
mean bubble radii of $15\,\mu$m for a pressure of $100\,$MPa ($\sim$ depth of $3.7\,$km) and of $30\,\mu$m for a pressure of $40\,$MPa ($\sim$ depth of $1.5\,$km). For a basaltic-andesitic magma containing larger amounts of water, the depth-size relation may differ. We nevertheless conclude that the tidal mechanism can significantly contribute to the bubble coalescence rate in the magma layers at one to several kilometres depth associated with bubble radii of $30 - 80\,\mu$m. In contrast, the tidal contribution gets negligible at shallow levels once the bubble radii are in the millimetre-range which corresponds to the bubble size range at which bubbles efficiently
start to segregate from the surrounding melt.

For a disperse bubble flow, the relative tidal contribution on the bubble coalescence rate depends reciprocally on the magma ascent rate, hardly on the gas volume fraction $\phi$, but approximately linearly on the volatile content $C_{H_2O}^0$ of the magma (see Figure 3). The tidal contribution causes an enhancement of the bubble coalescence rate equivalent to the enhancement caused





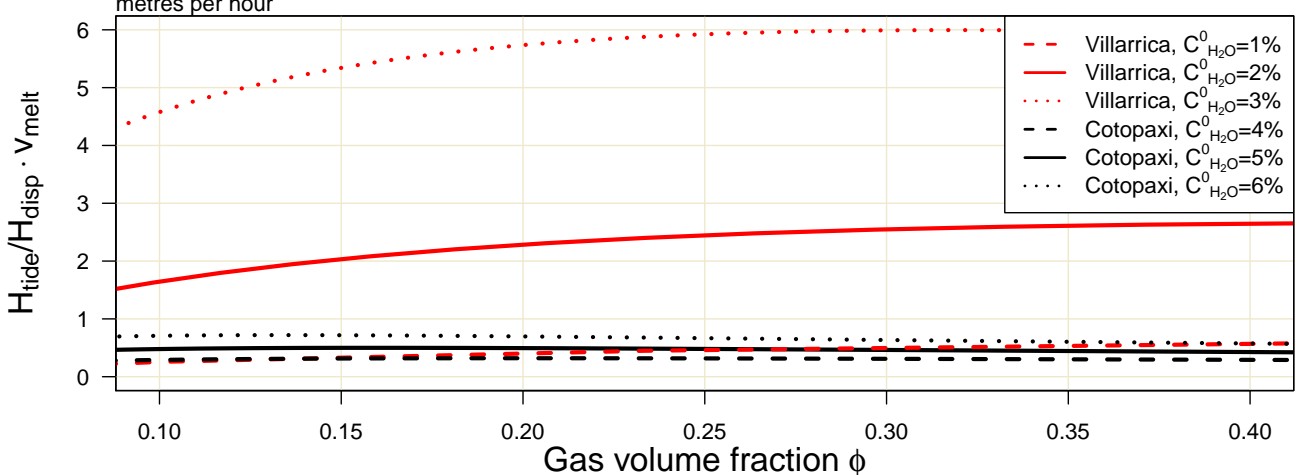

**Figure 3.** Relative contribution of the tidal mechanism (magnitude given by $H_{tide}$) on the bubble coalescence rate for a purely disperse bubble flow (magnitude given by $H_{disp}$) in dependency of the gas volume fraction and the initial water weight fraction of the magmatic melt. The corresponding values for $\phi$ are calculated with equation (5) assuming an ideal gas and magma temperatures of $1200°C$ for Villarrica and $1000°C$ for Cotopaxi. The relative tidal contribution is displayed as the equivalent to an enhancement of the magma ascent rate which would have the same effect on the bubble coalescence rate. The model increasingly loses validity above the percolation threshold of $\phi_{perc} \approx 0.3$.

by an increase of the magma ascent velocity by about $0.5\,\mathrm{m\,h^{-1}}$ for Cotopaxi and $2.5\,\mathrm{m\,h^{-1}}$ for Villarrica for the $C^0_{H_2O}$ listed in Table 1. For comparison, the magma ascent velocities in passively degassing volcanic systems vary roughly between $1-100\,\mathrm{m\,h^{-1}}$ (Cassidy et al., 2015, 2018). The tidal mechanism can accordingly contribute by at least several percent but potentially up to several ten percent to the overall bubble coalescence rate. For gas volume fractions exceeding the minimum percolation threshold of $\phi_{perc} \approx 0.3$, the model assumption of independent spherical bubbles increasingly loses its validity.

## 4   Discussion and Conclusions

Our model implies a tide-induced periodical vertical magma displacement in the conduit within every semi-diurnal cycle in the order of $0.1-1\,\mathrm{m}$ due to magma expansion in the reservoir. At Villarrica, the modelled vertical magma displacement of $0.45\,\mathrm{m}$ implies a periodic variation of the lava lake level (whose areal cross section is about 10 times larger than for the conduit, Goto and Johnson, 2011) of about $5\,\mathrm{cm}$. At Cotopaxi, the modelled vertical magma displacement of $0.09\,\mathrm{m}$ may apply additive stress on the solid plug.

We linked this magma displacement to bubble coalescence and compared the relative strength of the tide-induced bubble transport mechanism with respect to the classically predominant bubble transport mechanisms in magmas hosting a purely separated or a purely disperse or bubble flow. For both scenarios, we found that the tidal contributions to the overall bubble coalescence rate can be in the order of at least several percent up to several ten percent at a depth of several kilometres. At



shallower depth, the direct tide-induced contribution to the overall bubble coalescence rate are rather negligible because the classical transport mechanisms become more efficient.

The tide-enhanced bubble coalescence rate at greater depth can nevertheless affect the gas phase in the overlying shallower layer because the additionally coalesced bubbles have a larger buoyancy velocity as well as a reduced surface tension and can thus stimulate on the one hand enhanced volatile degassing from the melt phase to the gas phase and on the other hand enhanced bubble coalescence rates in overlying layers (Prousevitch et al., 1993). These enhancements can ultimately cause the percolation of the gas phase at a somewhat greater depth compared to the tide-free scenario. In consequence, the magma becomes gas-permeable at this greater depth potentially causing enhanced volcanic gas emissions (Rust and Cashman, 2011; Gonnermann et al., 2017). The additional contributions from this greater depth to the volcanic gas emissions may also slightly shift the chemical composition of the overall gas emissions towards the chemical composition of the gas phase at this greater depth when compared to the tide-free scenario (Burton et al., 2007).

The quantitative results have been derived for the tidal forcing during spring tide. In contrast, the amplitude of the tide-induced mechanism is smaller by a factor of 3 during neap tide. Accordingly, the amplitude of the additional tide-induced contributions to the coalescence rate varies within a spring-neap tide cycle entailing a periodical signal with a period of about 14.8 days superimposed on the (nevertheless potentially much stronger) tide-independent coalescence rate. For a disperse bubble flow scenario with rather fast magma ascent, a propagation of this superimposed signal from the enhanced coalescence rate via a variation of the percolation depth to the volcanic gas emissions is comprehensible. For a separated bubble flow scenario, however, the gas bubbles may need much more time than one spring-neap tide cycle to rise from a depth of several kilometres to the percolation depth. Magmatic systems can, however, become permeable already in a depth of $1 - 3\,\mathrm{km}$ (Edmonds and Gerlach, 2007; Burton et al., 2007), i.e. where the derived tidal effects are the strongest. In such a scenario, the tide-enhanced bubble coalescence rate could accordingly cause enhanced degassing without a significant delay.

In a scenario with a more shallow percolation depth, the periodic pattern could nevertheless propagate to the degassing signal because several crucial parameters such as the mean bubble radius $R_b$ and the gas volume fraction $\phi$ typically vary rather monotonously with pressure and thus depth (Gonnermann and Manga, 2013), implying a depth-dependency of the relative tidal contributions to the bubble coalescence rate. Convolved along the vertical conduit axis, the tide-enhanced coalescence rate may accordingly preserve an overall periodicity driven by the dominant contributions from those magma layers which are particularly sensitive to the tidal mechanism. Moreover, this pressure-dependency implies that gas contributions originating from the particularly tide-sensitive depths are more pronounced in the subsequent volcanic gas emissions during spring tide. Therefore tide-induced variations in the chemical composition within the volcanic gas plumes may particularly be manifested in the relative molar degassing ratios (e.g. Burton et al., 2007; Bobrowski and Giuffrida, 2012; Balcone-Boissard et al., 2016) associated with these depths.

In conclusion, we traced a possible tidal impact from the tidal potential to a magma expansion in the reservoir; to a vertical magma displacement profile in the conduit; to an enhanced bubble collision/coalescence rate; and ultimately motivated a link between the tide-enhanced bubble coalescence rate and the periodical signal in the observed volcanic gas emissions. Furthermore, we found plausible, by exemplary quantitative calculations, that the proposed tide-induced mechanism can lead to an enhancement of the bubble coalescence rate by up to several ten percent. If propagated from enhanced bubble coalescence to a variation in the magnitude or chemical composition of the volcanic gas emissions, a periodical spring tide





signal would be large enough to explain the observed about two-weekly variations in volcanic gas emissions.

Nevertheless, our conceptual model just aimed at a proof of concept. Future studies may increase the complexity of the model by e.g. (1) lifting several of our numerous simplifications (Appendix A), (2) incorporating macroscopic tidal mechanisms affecting the host rock explicitly, (3) adding several further microscopic mechanisms such as a tide-induced loosening of
bubbles attached to the conduit walls or the tidal impact on crystal orientation, and (4) investigating possible non-linear interferences between the tide-induced dynamics and the tide-independent magma convection flow.

*Data availability.* No unpublished data are presented or used.

*Acknowledgements.* We thank the Deutsche Forschungsgemeinschaft for supporting this work within the project DFG PL193/14-1.

## Appendix A: List of applied mayor simplifications

In our model we applied several simplifications regarding shape and physical properties of the magma plumbing system. This we did for the sake of clarity, and even more important, in order to isolate the tide-induced effect on magma flow and degassing. To achieve this, we (1) modelled the tide-induced magma flow in the conduit neglecting any tide-independent magma dynamics such as magma convection, which
implies an initial mechanical and thermodynamic equilibrium between magma and adjacent host rock. The only exception is the discussion of the impact of a constant magma ascent on the bubble coalescence rate. (2) Expansion of the initial conduit magma is neglected. (3) The host rock is assumed to be gas-tight. (4) Cylindrical volcanic conduit. (5) No-slip condition between conduit wall and magma. (6) Viscosity of the magma in the conduit assessed by the effective bulk viscosity. (7) Homogeneous magma flow properties. (8) Radial tide-induced magma displacement is neglected. Moreover, (9) bubble coalescence is modelled by bubble collision, neglecting near-field drainage processes, bubble
deformation processes, and post-collision coalescence processes. (10) Simple bubble size distributions are chosen, and (11) it is assumed that the volcanic gas phase exclusively consists of water vapour.

## Appendix B: Quantitative estimates for the geometrical model parameters

The conduit radius is a crucial model parameter. The uppermost 200 m of Villarrica's conduit frequently have been exposed during the decades prior to the 2015 eruption due to pronounced oscillations of the lava lake level (Moussallam et al., 2016; Johnson et al., 2018b). The
cross-sectional area of the conduit has a radius of about 30 m (Goto and Johnson, 2011) which at greater depths, however, narrows down to a mean radius of the order of $R_c = 6$ m as is implied by studies based on gas emission magnitudes (Palma et al., 2011) and seismo-acoustic properties (Richardson et al., 2014). The active vent of Cotopaxi was capped by an area of hot material with a diameter of 116-120 m during the eruption in 2015 (Johnson et al., 2018a). Although missing an empirical evidence, it is plausible that the mean conduit radius is somewhat narrower and therefore we assume a (rather conservative) value of $R_c = 40$ m.
Depth and volume of the magma reservoir constitute further crucial model parameters whose empirical estimates come with an even larger uncertainty. Seismic observations conducted at Villarrica imply the existence of a shallow magma reservoir with a lateral diameter of at least 5 km and a vertical extent of about 2.5 km whose centre of mass is located at a depth of around $D_r = 3$ km below the summit (Mora-Stock, 2015), implying a conduit length of about $L_c = 2$ km. Assuming an ellipsoidal magma reservoir this implies a magma reservoir volume of



$V_r = 35\,\text{km}^3$ at Villarrica. The magmatic system of Cotopaxi in contrast seems to be more complex and hosts a rather small magma pocket ($2\,\text{km}^3$) beneath the SW-flank and at a depth of about 4 km below the summit (Hickey et al., 2015). Seismic observations furthermore revealed fluid movements (magma and/or hydrothermal fluids) within a centrally located $85\,\text{km}^3$ column spanning from 2 to 14 km depth below the summit (Ruiz et al., 1998). This fluid column is assumed to connect the laterally offset shallow pocket with two much larger deeper magma

reservoirs, which are situated between 7-11 km and somewhere at a depth greater than 16 km below the summit, respectively (Arias et al., 2015; Mothes et al., 2017; Martel et al., 2018). For heating $85\,\text{km}^3$ of rock, these deep-seated magma reservoirs may be rather large. Missing any accurate volume estimate, we estimate that the upper of the two deep-seated reservoirs hosts a magma volume of $V_r = 35\,\text{km}^3$ with a centre of mass depth of $D_r = 8\,\text{km}$. The choice of equal reservoir volumes for both, Villarrica and Cotopaxi, allows for a better comparison of the impact of varying the other volcanic parameters. Further, we assume the small magma pocket as the lower end of the conduit, i.e. a

conduit length of $L_c = 4\,\text{km}$.

## Appendix C: Calculation of tide-induced conduit flow

**Oscillating centre of mass displacement** After a negligible settling time, the driven oscillator described by eq. 3 oscillates with semi-diurnal periodicity and we obtain the general long-term solution

$$
\begin{cases}
z(t) = z_0 \cdot \sin(\omega_{sd} \cdot t - \varphi_0) \\[2ex]
z_0 = \dfrac{a_0}{\sqrt{(\omega_0^2 - \omega_{sd}^2)^2 + (\gamma \cdot \omega_{sd})^2}} \\[2ex]
\varphi_0 = \arctan\left(\dfrac{\gamma \cdot \omega_{sd}}{\omega_0^2 - \omega_{sd}^2}\right)
\end{cases}
\tag{C1}
$$

**Navier-Stokes equation for periodical pipe flow** When exposed to a constant force (per unit mass) $f_{ext}^0$, a viscous fluid in a cylindrical pipe with radius $R_c$ flows with a parabolic velocity profile $v^0(r)$, $0 \leq r \leq R_c$,

$$
v^0(r) = \frac{R_c^2 \cdot f_{ext}^0}{4 \cdot \nu}\left[1 - \left(\frac{r}{R_c}\right)^2\right]
\tag{C2}
$$

When exposed to a periodically varying and thus time-dependent external force $f_{ext}(t) = f_{ext}^0 \cdot e^{i\omega t}$, the analytical solution of the flow profile is more complicated (Spurk, 1997)

$$
v(r,t) = \overline{v^0(r)} \cdot \Re\left[-i \cdot \frac{8}{N^2} \cdot e^{i\omega t} \cdot \left(1 - \frac{J_0(\sqrt{-i}\,N\,\frac{r}{R})}{J_0(\sqrt{-i}\,N)}\right)\right]
\tag{C3}
$$

with the centre of mass velocity $\overline{v^0(r)}$ of a constant forcing (see eq. C2), the real part $\Re[..]$, the imaginary unit $i$, the Bessel function $J_0(..)$, and the dimensionless parameter $N = \sqrt{\frac{\omega}{\nu}} \cdot R_c$. In the limit $N \to 0$, the velocity profile asymptotically adopts the time-dependency as well as the magnitude of the external force. For $N = 1$ the exact magnitude is already $0.98 \cdot f_{ext}^0$ and the radial profile shows hardly any deviation from a parabolic profile. For the chosen model parameters (Table 1) and $\omega = \omega_{sd}$, we obtain $N \approx 0.2$ and thus eq. C3 reduces in very good

approximation to the familiar

$$
v(r,t) \approx \frac{R_c^2 \cdot f_{ext}(t)}{4 \cdot \nu}\left[1 - \left(\frac{r}{R_c}\right)^2\right]
\tag{C4}
$$





**Derivation of the equation of motion (eq. 4)** The vertical velocity of the centre of mass can be obtained as $\dot{z}(t) = z_0 \cdot \omega_{sd} \cdot \cos(\omega_{sd} \cdot t - \varphi_0)$ from eq. (C1) and as $v(t) = (\pi \cdot R_c^2)^{-1} \cdot \int_0^{R_c} v(r,t) \cdot 2\pi r\, dr = \frac{R_c^2}{8 \cdot \nu} \cdot f_{ext}(t)$ from eq. (C4). Further, we know $f_{ext}(t) = f_{int}(t) = \gamma \cdot \dot{z}(t)$ from eq. (3). Applying $f_{ext}(t)$ to eq. (C4) reveals $\gamma = \frac{8 \cdot \nu}{R_c^2}$ and ultimately the fully parametrised equation of motion in eq. 4.

## Appendix D: Calculation of the collision volumes

As is common for most coalescence models (including those cited above), we consider spherical bubbles only. Two spherical bubbles with radii $f_1 \cdot R_b$ and $f_2 \cdot R_b$ ($f_1$ and $f_2$ drawn from $\delta_b^{size}(f)$) collides as soon as the distance between their bubble centres is $r_{coal} = (f_1 + f_2) \cdot R_b$. We introduce the "collision volume" $H(f_1, f_2; \Delta t)$ associated to a bubble with radius $f_1 \cdot R_b$ as the volume enclosing all possible initial locations of the bubble centre of another bubble with radius $f_2 \cdot R_b$ such that both bubbles collide/coalesce at the latest after a time interval $\Delta t$. All bubble collision mechanisms are derived as enhancements of the initial static collision volume

$$H_0(f_1, f_2) = \frac{4\pi}{3} \cdot R_b^3 \cdot (f_1 + f_2)^3 \tag{D1}$$

and we consider only those bubble pairs which have not collided already in the initial state. The absolute enhancement of the collision volume due to a particular bubble collision mechanism divided by $\Delta t$ thus gives the enhancement of the bubble collision rate contributed by the particular mechanism. Because the tide-induced mechanisms is derived for a semi-diurnal cycle, the relative strengths of all coalescence mechanisms are compared with respect to this time interval $\Delta t_{sd}$.

The collision volumes of the different collision mechanisms are all derived with the same approach: We fix the position of a bubble with arbitrary radius $f_1 \cdot R_b$ and derive $H(f_1, f_2; \Delta t)$ with respect to the relative motion of another bubble with arbitrary radius $f_2 \cdot R_b$. In each case the initial collision volume $H_0(f_1, f_2)$ is subtracted either already tacitly in the motivation or explicitly mathematically. Higher-order details such as the influence of a third bubble on the numeric results are ignored.

**Tide-enhanced bubble collision volume** We fix the horizontal coordinates $(r, \varphi)_{bubble1} = (r_0, 0)$, $0 \le r_0 \le R_c$, of the first bubble, where the cylindrical symmetry of the conduit allows to pick the azimuth angle without loss of generality, and vary the horizontal coordinates $(r, \varphi)_{bubble2} = (r, \varphi)$ of a second bubbles. The horizontal distance $h$ between the two bubbles is thus given by $r^2 = r_0^2 - 2 \cdot r_0 \cdot h \cdot \cos(\varphi) + h^2$. Within a semi-diurnal cycle, the peak-to-peak differential tide-induced vertical displacement of two bubbles at the radial coordinates $r$ and $r_0$ is given by $\Delta z_{tide}(r, r_0) = 2 \cdot |z_0(r) - z_0(r_0)|$ (see eq. 4). The tide-induced collision volume is then the integral of $\Delta z_{tide}(r, r_0)$

integrated over a circle with radius $r_{coal}$:

$$H_{tide}(r_0) = \int_0^{r_{coal}} dh\, h \int_0^{2\pi} d\varphi\, \Delta z_{tide}(r, r_0) \tag{D2}$$

$$= \frac{4\,\Psi\,r_0}{R_c^2} \int_0^{r_{coal}} dh\, h^2 \int_0^{2\pi} d\varphi \left| \cos(\varphi) - \frac{h}{2\,r_0} \right| \tag{D3}$$





This integral has to be split in two integrals at the angles where the sign of the absolute function changes, which is the case at $\pm\varphi' = \pm\arccos(\frac{h}{2\,r_0}) \approx \pm\frac{\pi}{2}$:

$$H_{tide}(r_0) = \frac{16\,\Psi\,r_0}{R_c^2} \int\limits_0^{r_{coal}} dh\, h^2 \underbrace{\left[\sin(\varphi') - \cos(\varphi')\cdot\varphi'\right]}_{\approx 1\text{ for }h<<r_0} \tag{D4}$$

$$\approx \frac{16\,\Psi\,r_0}{R_c^2}\cdot\frac{r_{coal}^3}{3} \tag{D5}$$

$$= \frac{4\,\Psi\,r_0}{\pi\,R_c^2}\cdot H_0(f_1,f_2) \tag{D6}$$

We integrate $H_{tide}(r_0)$ over the local spatial bubble distribution in the conduit in order to obtain the average effect. We parametrise the (isotropic) spatial bubble distribution by the depth-independent $\delta_b^{spatial}(r_0) = (1+\alpha)\cdot\frac{1}{R}\cdot(\frac{r_0}{R})^\alpha$, which is an homogeneous distribution for $\alpha = 1$ but with all bubbles at the conduit wall if $\alpha \to \infty$, respectively. We obtain for the averaged tide-induced collision volume

$$H_{tide} = \int\limits_0^R \sigma_{tide}(r_0)\cdot\delta_b^{spatial}(r_0)\cdot dr_0 \tag{D7}$$

$$= \underbrace{\left[\frac{1+\alpha}{2+\alpha}\right]}_{\text{distribution}}\cdot\underbrace{\left[\frac{4\cdot\Psi}{\pi\cdot R_c}\right]}_{\text{tidal}}\cdot\underbrace{H_0(f_1,f_2)}_{\text{scale}} \tag{D8}$$

The "distribution term" is $\frac{2}{3}$ for an isotropic bubble distribution and approaches unity if all bubbles are close to the host rock. Arguably, the conditions for crystal nucleation and thus bubble nucleation are better close to the host rock where the magma is cooler and more crystals and thus nucleation possibilities are available. Following this reasoning but also because we want to examine the maximum possible tidal impact, we set the distribution term to unity. The "tidal term" contains the information on the scale of the effective tide-induced impact. The "scale term" contains the information on the actual bubble size distribution, highlighting that the relative tidal enhancement is identical for any bubble size distribution, at least in our simple model.

**Buoyancy-induced bubble collision volume** Two bubbles with radii $f_1\cdot R_b \neq f_2\cdot R_b$ have a differential rise velocity $\Delta v_{buoy} = |f_2^2 - f_1^2|\cdot v_{buoy}(R_b)$ and thus their relative distance changes during the rise. The two bubbles will collide if the larger bubble is below the smaller and if the horizontal distance between their bubble centres is at most $r_{coal}$. Accordingly, the buoyancy-induced collision volume $H_{buoy}$ is a cylindrical volume with base area $\pi\cdot r_{coal}^2$ and cylinder length $\Delta v_{buoy}\cdot\Delta t_{sd}$:

$$H_{buoy}(f_1,f_2) = \pi\cdot r_{coal}^2\cdot|f_2^2 - f_1^2|\cdot v_{buoy}(R_b)\cdot\Delta t_{sd} \tag{D9}$$

$$= \frac{3\cdot|f_2 - f_1|}{4\cdot R_b}\cdot v_{buoy}(R_b)\cdot\Delta t_{sd}\cdot H_0(f_1,f_2) \tag{D10}$$

For a given pair of bubbles with radii $f_1\cdot R_b \neq f_2\cdot R_b$, $f_1$ and $f_2$ drawn from $\delta_b^{size}(f)$, the ratio of the contribution from the tide-induced and the buoyancy-induced collision mechanisms is

$$\frac{H_{tide}}{H_{buoy}} = \frac{24\cdot\Psi\cdot\nu}{\pi\cdot R_{ic}\cdot|f_1 - f_2|\cdot g\cdot R_b\cdot\Delta t_{sd}} \tag{D11}$$

The bulk ratio (with respect to the local magma layer) can be obtained by a previous and separate integration of $H_{tide}$ and $H_{buoy}$ over $f_1$ and $f_2$ with respect to the actual bubble size distribution $\delta_b^{size}(f)$ (rather than integrating eq. D11). For the explicit bubble size distribution





$\tilde{\delta}_b^{size}$ from eq. 6, we obtain the bulk collision volumes $\tilde{H}_{tide}$ and $\tilde{H}_{buoy}$

$$\frac{\tilde{H}_{tide}(q)}{H_0(1,1)} = (1 + 0.89 \cdot q + 0.11 \cdot q^2) \cdot \frac{4 \cdot \Psi}{\pi \cdot R_c} \tag{D12}$$

$$\frac{\tilde{H}_{buoy}(q)}{H_0(1,1)} = (q - q^2) \cdot \frac{9}{16 \cdot R_b} \cdot v_{buoy}(R_b) \cdot \Delta t_{sd} \tag{D13}$$

and thus the bulk ratio (used for the calculation of Figure 2)

$$\frac{\tilde{H}_{tide}}{\tilde{H}_{buoy}} = 60 \cdot \left(0.9 + \frac{1+q^2}{q-q^2}\right) \cdot \frac{\nu[\mathrm{m^2\,s^{-1}}] \cdot \Psi[\mathrm{m}]}{R_{ic}[\mathrm{m}] \cdot R_b[\mathrm{\mu m}]} \tag{D14}$$

**Growth-induced bubble collision volume** In magma with a disperse bubble flow ($v_{buoy} << v_{melt}$), a rising bubble exhibits a pressure decrease rate by

$$\frac{\Delta p}{\Delta t} = v_{melt} \cdot (\nabla p)_{vert} \tag{D15}$$

Ignoring accompanying changes in secondary parameters such as melt temperature and magma composition and assuming for simplicity a monodisperse bubble size distribution (thus $R_b^3 \propto \phi$), we obtain for the enhancement of the collision volume due to a rise-driven pressure decrease by $\Delta p << p_0$ (apply eq. 5 on eq. D1)

$$H_{disp}(\Delta p; p_0) = H_0(R_b(p_0 - \Delta p)) - H_0(R_b(p_0))$$

$$= H_0(1,1) \cdot \frac{C_{H_2O}^0 - \frac{1}{2}\sqrt{K_{H_2O} \cdot p_0}}{C_{H_2O}^0 - \sqrt{K_{H_2O} \cdot p_0}} \cdot \frac{\Delta p}{p_0} + \mathcal{O}\left[\left(\frac{\Delta p}{p_0}\right)^2\right] \tag{D16}$$

where we assumed that $\rho_{melt}$ is constant and $\rho_{gas}$ follows the ideal gas law. Inserting eq. D15 in eq. D16, we obtain:

$$\frac{H_{disp}(p_0)}{H_0(1,1)} = \frac{C_{H_2O}^0 - \frac{1}{2}\sqrt{K_{H_2O} \cdot p_0}}{C_{H_2O}^0 - \sqrt{K_{H_2O} \cdot p_0}} \cdot v_{melt} \cdot \Delta t_{sd} \cdot \frac{(\nabla p)_{vert}}{p_0} \tag{D17}$$

The ratio of the contribution from the tide-induced and the growth-induced collision mechanism (used for the calculation of Figure 3) is

$$\frac{H_{tide}}{H_{disp}} = \underbrace{\frac{C_{H_2O}^0 - \sqrt{K_{H_2O} \cdot p_0}}{C_{H_2O}^0 - \frac{1}{2}\sqrt{K_{H_2O} \cdot p_0}}}_{\approx 0.25 - 0.5} \cdot \frac{4 \cdot \Psi[\mathrm{m}] \cdot p_0\,[\mathrm{MPa}]}{R_c[\mathrm{m}] \cdot v_{melt}[\mathrm{m\,h^{-1}}]} \tag{D18}$$



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
