# Peer review of "On the link between Earth tides and volcanic degassing"

_Solid Earth, 2019_

## Referee Comment (RC1) · Anonymous Referee #1 · 11 Mar 2019

I revised the manuscript "On the link between Earth tides and volcanic degassing" by Dinger et al., submitted to Solid Earth (EGU).

This is an interesting paper. The manuscript reports a model of a simplified magmatic system response to tidal stresses. The model provides evidences that the effects of periodical signals (i.e., tidal forces) may enhance bubble coalescence in the magma thus explaining periodical changes in volcanic gas emissions.

The paper is clear, well written and right to the point. The subject is of general interest and, as far as I am aware, no significant portions of the manuscript have been published elsewhere.

The manuscript needs minor revisions before publication. Specifically, I have the fol-

lowing main area of concern with the manuscript:

General comments Assumptions of the model: There is ample evidence that the transition from an explosive eruption regime to an effusive regime can be due to magma losing gas to fractured country rock during storage and/or ascent towards the surface. As a consequence, the gas volume fraction exhibits complex variations with height and with horizontal distance from the walls, which is not compatible with closed system degassing. I believe that the here proposed model tacitly assumes that 1) tidal stresses do not affect the wall-rock permeability and 2) tidally induced gas loss from wall rocks cannot explain the observed periodic variations of volcanic degassing. Maybe, these points should be explicitly discussed (see also Manga et al., 2012 Changes in permeability caused by transient stresses: Field observations, experiments, and mechanisms, Reviews of Geophysics 50 (2)).

- the max. vertical tidal acceleration a0 reported in table 1 has the dimension of a velocity. Any consequences on the reliability of the model? Just a typing error?

---

## Referee Comment (RC2) · Nolwenn Le Gall (Referee) · 1 Apr 2019

In this paper, the authors present a model quantifying the impact of Earth tides on volcanic gas emissions, by enhancing bubble coalescence. This model has the merit of being the first of its kind. The paper is well written and the model clearly exposed, although I have made some comments and minor technical corrections, see below.

Gereral comments: (1) While clearly exposed in the text and in Appendix A, the major simplifications of the model make its applicability questionable. (2) It would be good to add a figure on the flux of volcanic gas emissions and its variation regarding tidal pattern, as well as to give some numbers (eg measured volcanic gas ratios) in the text. This would better illustrate the periodic tidal impact on volcanic degassing, which is at

[Figure]

the core of this paper.

Specific comments: (1) The locations of the two volcanoes chosen to illustrate the model could be added in the text for information, lines 23-24 p.3. (2) Why using the solubility model of Liu et al. (2005) determined for rhyolitic melts, as your model deals with basaltic and andesitic compositions (eg Table 1)? Similarly, why introducing a comparison with a rhyolitic composition (lines 9-10 p.10)? (3) Is there any order of magnitude that could be given for magma displacement, line 20 p.7? (4) Lines 22-23 and 30 p.8. In natural magmas, bubble sizes can also follow exponential and mixed exponential–power law distributions. I would tend to suggest that in the case of your model, considering equilibrium degassing and the importance of bubble coalescence, the best estimate of the bubble size distribution may not be a power law (eg Le Gall and Pichavant, 2016, JVGR). (5) Line 6 p.9. Diffusion-driven volatile degassing could also take place in contact with the host rock. The volatiles could be lost from the magmatic melt by diffusion.

Technical corrections: (1) Delete vapour line 10 p.8, as you are talking about the melt phase (with dissolved volatiles) and not the gas phase. (2) You could also delete the word vapour line 12 p.8. (3) Line 5 p.10, the brackets can be deleted.

---

## Author Comment (AC1) · 15 Apr 2019

**Anonymous Reviewer #1, Heading Paragraphs: This is an interesting paper. The manuscript reports a model of a simplified magmatic system response to tidal stresses. The model provides compelling evidences that the effects of periodical signals (i.e., tidal forces) may enhance bubble coalescence in the magma thus explaining periodical changes in volcanic gas emissions. The paper is clear, well written and right to the point. The subject is of general interest and, as far as I am aware, no significant portions of the manuscript have been published elsewhere.**

**Nolwenn Le Gall, Heading Paragraphs: In this review, the authors present a model quantifying the impact of Earth tides on volcanic gas emissions, by**

[Figure]

**enhancing bubble coalescence. This model has the merit of being the first of its kind. The paper is well written and the model clearly exposed, although I have made some comments and minor technical corrections, see below.**

We thank the Anonymous Reviewer #1 and Nolwenn Le Gall for highlighting the relevance and in particular the clear style of the manuscript. It was a major intention to keep it that clear in order to provide a well-prepared foundation for the subsequent tackling of some potentially too oversimplified assumptions. We also want to thank for pointing out some minor inaccuracies in the submitted manuscript. We are convinced that we were able to answer their comments comprehensibly and comprehensively either by specifying the marked paragraphs on the manuscript or by respectfully rebutting.

All of our references to pages and lines refer to the new version of the manuscript.

**Anonymous Reviewer #1, General Comment: the here proposed model tacitly assumes that 1) tidal stresses do not affect the wall-rock permeability...** We indeed ignored any tide-induced variations of the host rock permeability in our model. This has been stated explicitly in the Appendix A (*"(3) The host rock is assumed to be gas-tight"*, page 13, lines 16-17) and also in the conclusions *"(2) incorporating macroscopic tidal mechanisms affecting the host rock explicitly, (3) adding several further microscopic mechanisms such as a tide-induced loosening of bubbles attached to the conduit walls"*, page 13, lines 3-5). We admit that an earlier and more to the point mentioning of this simplification is appropriate and thus added some sentences.
**Change:** We added (1) *a variation of the host rock permeability (Bower, 1983; Elkhoury et al., 2006; Manga et al., 2012),* on page 2 lines 16-17 and (2) *Furthermore, the tide could also cause a variation of the host rock permeability (Bower, 1983; Elkhoury et al., 2006; Manga et al., 2012). This mechanism and its possible interference with the here presented concept is ignored in our model.* on page 5, lines 11-13.

**Anonymous Reviewer #1, General comment (continued): ... and 2) tidally induced gas loss from the wall rocks cannot explain the observed periodic variations of the volcanic degassing.** There is no evidence/quantitative estimate available that the periodic variations in the volcanic degassing can be explained by wall rock effects alone. In particular, there is no evidence that no other tide-induced mechanisms can act simultaneously. Our manuscript can give rise for a quantitative comparison once such a quantitative model has been established.

**Anonymous Reviewer #1, Minor correction: the max. vertical tidal acceleration a0 reported in table 1 has the dimension of a velocity. Any consequences on the reliability of the model? Just a typing error?**
**Change:** We corrected the typo. No consequence for the numerical model output.

---

## Author Comment (AC2) · 15 Apr 2019

**Nolwenn Le Gall, Heading Paragraphs: In this review, the authors present a model quantifying the impact of Earth tides on volcanic gas emissions, by enhancing bubble coalescence. This model has the merit of being the first of its kind. The paper is well written and the model clearly exposed, although I have made some comments and minor technical corrections, see below.**

**Anonymous Reviewer #1, Heading Paragraphs: This is an interesting paper. The manuscript reports a model of a simplified magmatic system response to tidal stresses. The model provides compelling evidences that the effects of periodical signals (i.e., tidal forces) may enhance bubble coalescence in the**

[Figure]

**magma thus explaining periodical changes in volcanic gas emissions. The paper is clear, well written and right to the point. The subject is of general interest and, as far as I am aware, no significant portions of the manuscript have been published elsewhere.**

We thank Nolwenn Le Gall and the Anonymous Reviewer #1 for highlighting the relevance and in particular the clear style of the manuscript. It was a major intention to keep it that clear in order to provide a well-prepared foundation for the subsequent tackling of some potentially too oversimplified assumptions. We also want to thank for pointing out some minor inaccuracies in the submitted manuscript. We are convinced that we were able to answer their comments comprehensibly and comprehensively either by specifying the marked paragraphs on the manuscript or by respectfully rebutting.

All of our references to pages and lines refer to the new version of the manuscript.

**Nolwenn Le Gall, General Comment (1): While clearly exposed in the text and in Appendix A, the major simplifications of the model make its applicability questionable.** We do not claim to present an ultimate model of the tidal impact on volcanic degassing. In fact, we highlighted in the conclusions *"Nevertheless, our conceptual model just aimed at a proof of concept. Future studies may increase the complexity of the model by e.g. (1) lifting several of our numerous simplifications [...]"* (page 13, lines 2-3).

**Nolwenn Le Gall, General Comment (2): It would be good to add a figure on the flux of volcanic gas emissions and its variation regarding tidal pattern, as well as to give some numbers (eg measured volcanic gas ratios) in the text. This would better illustrate the periodic tidal impact on volcanic degassing, which is at the core of this paper.** Figures on the evolution and propagation of volcanic

gas emissions can be found in the literature. Furthermore, we actually do not model the behaviour of the gas fluxes but "only" the tidal impact on the bubble coalescence rate. Adding such a figure may imply that the model does also explicitly model the gas fluxes.

We hesitate to cite the absolute numbers of the tidal signals reported in the literature on volcanic degassing because: (1) The crucial empirical findings are the observations of persistent and significant periodical degassing signals but not their mere variations of the absolute numbers. Nevertheless, giving e.g. the relative ratio between the amplitude of the periodic signals and the "background base-line" could be helpful indeed. But (2) these numbers are only of limited use for the assessment of the empirical evidence, in particular when cited without a further context. And (3) adding these numbers (or even the elsewhere discussed context) would reduce the readability of the manuscript. Considering these three arguments, we decided that giving the interested reader a comprehensive list of the literature at hand is the more appropriate approach.

**Nolwenn Le Gall, Specific Comment (1): The location of the two volcanoes chosen to illustrate the model could be added in the text for infomation.**
**Change:** We added their latitudes to the main text which now reads *"mid-latitude Villarrica (39.5° S) and equatorial Cotopaxi (0.7° S) volcanoes"* (page 3, lines 22-23).

**Nolwenn Le Gall, Specific Comment (2): Why using the solubility model of Liu et al. (2005) determined for rhyolitic melts, as your model deals with basaltic and andesitic compositions (eg Table 1)? [...]** We use the most simple approximation of the (entire) volatile solubility, namely $\sim \sqrt{K_{H_2O} \cdot p}$. We are aware that $K_{H_2O}$ varies with temperature and magma composition. In our model, a modest variation of $K_{H_2O}$ has only a rather small quantitative effect on the model output (see equation (D18)). Therefore, we decided to keep the modelling of $K_{H_2O}$ as simple as possible because a more complex modelling would hardly improve the quantitative

model estimates but disproportionally reduce the readability of our proof of concept. Accordingly, we (tacitly) ignored this natural range of $K_{H_2O}$ and chose a plausible "mean value" of $10^{-11}\,Pa^{-1}$ in order to retrieve the quantitative model estimates.

We referred to Liu et al. (2005) because they proposed for the first time (to our knowledge) a sufficiently simple empirical formulation for $K_{H_2O}$. Zhang et al. (2007) have reported that the model from Liu et al. (2005) is also approximately valid for basaltic and inter-mediate melt. In particular, the chosen value of $10^{-11}\,Pa^{-1}$ appears to be also a valid estimate for the solubility in basaltic/andesitic magmas. We admit that referring to Zheng et al. (2007) is more appropriate.

**Change:** We changed the reference from Liu et al. (2015) to Zhang et al. (2007) on page 8, lines 16-17 and in Table 1.

**Nolwenn Le Gall, Specific Comment (2) (continued): [...] Similarly, why introducing a comparison with a rhyolitic composition (lines 9-10 p.10)?** There we aimed to give a quantitative scale for the bubble size as a function of the pressure. We have not found comparably clear data for basaltic/andesitic magma which gave the dependency of the bubble size on the pressure. We highlighted our awareness that the quantitative scale may differ for other magma: *For a basaltic-andesitic magma containing larger amounts of water, the depth-size relation may differ* (page 10, line 11). Le Gall and Pichavant (2016) reports such depth-size relations for basaltic magma. We were not aware of this (and apparently other related) publications and thankfully incorporate its findings in our manuscript. We want to highlight that out findings derived from Figure 2 still holds true not only qualitatively but also similarly quantitatively.

**Change:** We specified the paragraph on depth-size relation: *"For comparison, Le Gall and Pichavant (2016) obtained from basalt decompression experiments mean bubble radii of (at most, depending on the volatile content) 23 µm for a pressure of 100 MPa (∼ depth of 3.7 km) and of 80 µm for a pressure of 50 MPa (∼ depth of 1.9 km) and concluded an extensive bubble coalescence rate at depth associated whit*

[Figure]

$50 - 100$ *MPa. Similarily, Castro et al. (2012) obtained from rhyolite decompression experiments mean bubble radii of 15 µm for a pressure of 100 MPa ($\sim$ depth of 3.7 km) and of 30 µm for a pressure of 40 MPa ($\sim$ depth of 1.5 km). For andesitic magma, the dependency of the bubble size on the pressure is presumably between the values for the basaltic and the rhyolitic magma. We conclude that the tidal mechanism can significantly contribute to the bubble coalescence rate in magma layers at a depth greater than one kilometre, associated with bubble radii of $30 - 80 \, m$"* (page 10, lines 9-16).

**Nolwenn Le Gall, Specific Comment (3): Is there any order of magnitude that could be given for magma displacement, line 20 p.7?** We are not sure whether we understand the question correctly but assume that it refers to the maximum magma displacement occurring during a semi-diurnal cycle. The maximum magma displacement is given by $\Psi$, see equation (4). The maximum magma displacement is the largest during spring tide, i.e. $\Psi$ is slowly time-dependent. We gave quantitative values $\Psi_{vill}^{st}$ and $\Psi_{coto}^{st}$ during spring tide for both model volcanoes on page 13, line 13/15.

**Nolwenn Le Gall, Specific Comment (4): Lines 22-23 and 30 p.8. In natural magmas, bubble sizes can also follow exponential and mixed exponential-power law distributions. I would tend to suggest that in the case of your model, considering equilibrium degassing and the importance of bubble coalescence, the best estimate of the bubble size distribution may not be a power law (eg Le Gall and Pichavant, 2016, JVGR).** We were not aware of this publication. We changed our general statements in order to incorporate these new findings on the bubble size distributions. These changes have nevertheless no effect on our model.
**Change:** We changed the general statement on polydisperse bubble size distributions to *"Bubbles typically vary in size following a power law (Cashman and Marsh, 1988, Blower et al., 2003) or a mixed power-law exponential distribution (Le Gall and ,*

*2016)..."* (page 8, lines 22-24). Furthermore, we accordingly adapted the motivation for the simplified polydisperse bubble size distribution used in our model to *"An estimate of a power-law bubble size distribution would require three parameters: the exponent and the lower and upper truncation cut-off (Lovejoy et al., 2004). An estimate of a mixed power-law exponential bubble size distribution would require at least two further parameters. The following analysis is conducted for an arbitrary bubble size distribution, nevertheless, for a basic quantitative estimate, we mimic a proper polydisperse bubble size distribution by the simpler single-parametric..."* (page 8, line 26ff.)

**Nolwenn Le Gall, Specific Comment (5): Line 6 p.9. Diffusion-driven volatile degassing could also take place in contact with the host rock. The volatiles could be lost from the magmatic melt by diffusion.** We agree with the reviewer that this could be possible. This could have some relevance for the actual statement on page 9, line 6 as well as for a (more comprehensive) model. However, diffusion of volatiles in magmatic melts may be considered to be a quite slow process that rather acts on microscopical distances at the timescales we consider here (e.g. Freda et al., 2005, *Sulfur diffusion in basaltic melts*). We thus assume that diffusion hardly can bridge larger than microscopic distances, and therefore would merely be effective close to the conduit walls. Thus it is highly questionable that diffusion would cause a considerable change in the volcanic outgassing signal measured at the surface of a volcano, as long as other much more efficient gas transport mechanisms are acting.
*On the specific statement:* We argue that the meaning/validity of the statement is clear in the context as how we introduced the model strategy, i.e. we announced to investigates the effect of the tide-induced radial displacement profile.
*On the model in general:* Firstly, host-rock-facilitated volatile degassing/bubble nucleation/bubble coalescence is just an additional mechanisms besides the introduced coalescence mechanisms for a "separated" or "dispersed" bubble flow. Neglecting this mechanism could underestimate the "classical" bubble coalescence rate and vice

versa overestimate the tide-induced enhancement of the bubble coalescence rate. We find it nevertheless plausible that the vast majority of the coalescence events is caused by the two mentioned classical mechanism and thus this overestimation of the tidal contribution would be much smaller than other uncertainties caused by the some of the applied simplifications. Secondly (and see our reply on the referee report #1), it has been stated explicitly in the Appendix A that (*"(3) The host rock is assumed to be gas-tight"*, page 13, lines 16-17) and also in the conclusions *"(2) incorporating macroscopic tidal mechanisms affecting the host rock explicitly, (3) adding several further microscopic mechanisms such as a tide-induced loosening of bubbles attached to the conduit walls"*, page 13, lines 3-5). We admit that an earlier and more to the point mentioning of this simplification is appropriate and thus added some sentences.
**Change:** We added (1) *"a variation of the host rock permeability (Bower, 1983; Elkhoury et al., 2006; Manga et al., 2012),"* on page 2 lines 16-17 and (2) *"Furthermore, the tide could also cause a variation of the host rock permeability (Bower, 1983; Elkhoury et al., 2006; Manga et al., 2012). This mechanism and its possible interference with the here presented concept is ignored in our model"* on page 5, lines 11-13.

**Nolwenn Le Gall, Technical corrections: (1) Delete vapour line 10 p.8, as you are talking about the melt phase (with dissolved volatiles) and not the gas phase. (2) You could also delete the word vapour line 12 p.8. (3) Line 5 p.10, the brackets can be deleted.** We agree.
**Change:** We applied all proposed technical corrections. Specifically, correction (3) now reads *"A more comprehensive formulation of the classically predominant bubble transport/approaching mechanisms has been proposed, e.g., by Mancini et al. (2016)"*.

**Additional technical corrections:** (1) We distinguished between a "separated" and "disperse" bubble flow. The latter is actually called a "*dispersed* bubble flow" (see e.g. Gonnermann and Manga, 2012). We corrected this term everywhere throughout

the manuscript. (2) The "chosen reference value" $\phi = 0.2$ in Table 1 is an historical artefact not further used in the manuscript (Figure 3 shows results for a range of $\phi$). Instead, we highlighted the general constraint $\phi < \phi_{perc}$ also in Table 1.
* * *